



**Greenhouse gas production in degrading ice-rich permafrost deposits in northeast Siberia**
Josefine Walz[1,2], Christian Knoblauch[1,2], Ronja Tigges[1], Thomas Opel[3,4], Lutz Schirrmeister[4], Eva-Maria
Pfeiffer[1,2]
[1]Institute of Soil Science, Universität Hamburg, Hamburg, 20146, Germany
[2]Center for Earth System Research and Sustainability, Universität Hamburg, Hamburg, 20146, Germany
[3]Permafrost Laboratory, Department of Geography, University of Sussex, Brighton, BN1 9RH, UK
[4]Alfred Wegener Institute Helmholtz Centre for Polar and Marine Research, Periglacial Research
Section, 14473 Potsdam, Germany
*Correspondence to*: Josefine Walz (josefine.walz@uni-hamburg.de)
**Abstract**
Permafrost deposits have been a sink for atmospheric carbon for millennia. Thaw-erosional processes,
however, can lead to rapid degradation of ice-rich permafrost and the release of substantial amounts of
organic carbon (OC). The amount of the OC stored in these deposits and their potential to be microbially
decomposed to the greenhouse gases carbon dioxide ($CO_2$) and methane ($CH_4$) depends on climatic
and environmental conditions during deposition and the decomposition history before incorporation into
the permafrost. Here, we examine potential greenhouse gas production in degrading ice-rich permafrost
deposits from three locations in the northeast Siberian Laptev Sea region. The deposits span a period
of about 55 kyr and include deposits from the last glacial and Holocene interglacial periods. Samples
from all three locations were aerobically and anaerobically incubated for 134 days at 4 °C. Greenhouse
gas production was generally higher in glacial than Holocene deposits. In permafrost deposits from the
Holocene and the late glacial transition, only 0.1–4.0% of the initially available OC could be decomposed
to $CO_2$, while 0.2–6.1% could be decomposed in glacial deposits. Within the glacial deposits from the
Kargin interstadial period (Marine Isotope Stage 3), local depositional environments, especially soil
moisture, also affected the preservation of OC. Sediments deposited under wet conditions contained
more labile OC and thus produced more greenhouse gases than sediments deposited under drier
conditions. To assess the long-term production potentials, deposits from two locations were incubated
for a total of 785 days. However, more than 50% of the aerobically produced and more than 80% of
anaerobically produced $CO_2$ after 785 days of incubation were already produced within the first 134 days,
highlighting the quantitative importance of the slowly decomposing OC pool in permafrost. $CH_4$



production was generally observed in active layer samples but only sporadically in permafrost samples and was several orders of magnitude smaller than $CO_2$ production.

Key words: Permafrost carbon, greenhouse gases, incubation, Yedoma, Siberian Arctic

## 1    Introduction

Permafrost, i.e. ground that is at ≤0 °C for at least two consecutive years (van Everdingen, 2005), may preserve organic matter (OM) for millennia (Ping et al., 2015). The current organic carbon (OC) pool of soils and sediments in permafrost-affected landscapes is estimated to be ~1300 Pg, of which ~800 Pg are perennially frozen (Hugelius et al., 2014). However, warming-induced environmental changes and permafrost degradation could lead to rapid thaw of substantial amounts of currently frozen OM, microbial decomposition of the thawed material, and rising greenhouse gas fluxes to the atmosphere (Natali et al., 2015; Schuur et al., 2015). The changes are expected to be most pronounced in near-surface layers (Schneider von Deimling et al., 2012), but thermo-erosion of ice-rich permafrost, i.e. permafrost with >20 vol% ice (Brown et al., 1998), also enables deep thaw of several tens of meters (Schneider von Deimling et al., 2015).

Ice-rich permafrost deposits, also called ice complex deposits, accumulated in unglaciated Arctic lowlands. During cold stages, fine grained organic-rich material of polygenetic origin was deposited on predominantly flat plains (Schirrmeister et al., 2013). The deposits are dissected by large ice wedges, which can amount for up to 60vol% (Ulrich et al., 2014). The most prominent ice complex deposits, referred to as Yedoma, accumulated during the late Pleistocene between >55 and 13 ka before present (BP), i.e. during the Marine Isotope Stages (MIS) 3 and 2 (Schirrmeister et al., 2011). Locally, however, remnants of older ice complex deposits of MIS 7 or MIS 5 age are also preserved (Opel et al., 2017; Schirrmeister et al., 2002; Wetterich et al., 2016).

The thickness of Yedoma deposits in Siberia (Grosse et al., 2013) and Alaska (Kanevskiy et al., 2011) can reach >50 m. At the time of deposition; rapid sedimentation and freezing incorporated relatively undecomposed OM into the permafrost (Strauss et al., 2017). However, owing to the high ice content, Yedoma deposits are highly susceptible to warming-induced environmental changes, erosion, and ground subsidence following permafrost thaw. Only 30% of the Yedoma region (~416,000 km$^2$) is considered intact (Strauss et al., 2013) while the other 70% have already undergone some level of permafrost degradation (e.g. Morgenstern et al., 2013). Today, the whole Yedoma region stores 213–



456 Pg of OC, of which 83–269 Pg are stored in intact Yedoma and 169–240 Pg in thermokarst and refrozen taberal deposits (Hugelius et al., 2014; Strauss et al., 2013, 2017; Walter Anthony et al., 2014; Zimov et al., 2006). But, high spatial and temporal variability result in large uncertainties of how much OC is thawed out from degrading ice-rich permafrost deposits and how much of this OC can be microbially decomposed to carbon dioxide ($CO_2$) or methane ($CH_4$) after thaw.

In addition to the quantity of OM, its decomposability will influence how fast the OC in Yedoma deposits can be transformed into $CO_2$ or $CH_4$ (Knoblauch et al., 2018; MacDougall and Knutti, 2016). Since plants are the main source of OM in soils, vegetation composition plays an important role for OM decomposability (Iversen et al., 2015). Furthermore, OM has undergone different degradation processes before being incorporated into permafrost depending on permafrost formation pathways (Harden et al., 2012; Waldrop et al., 2010). In epigenetic permafrost, that is permafrost aggradation through intermittent freezing after the material was deposited, OM has already undergone some level of transformation and easily decomposable, labile OC compounds are decomposed and lost to the atmosphere prior to incorporation into the permafrost (Hugelius et al., 2012). In contrast, OM in syngenetically frozen Yedoma, i.e. concurrent material deposition and permafrost aggregation, had little time to be transformed prior to freezing and may thus contain high amounts of labile OM, which may be quickly decomposed to greenhouse gases after thaw (Dutta et al., 2006). In this case, the amount and decomposability of the fossil OM is controlled by the OM source, i.e. predominantly vegetation, which in turn depends on paleo-climatic conditions (Andreev et al., 2011).

The decomposability of permafrost OM is often assessed based on OM degradation proxies, total OC (TOC) content, total organic carbon to- total nitrogen ratios (C/N), or stable carbon isotopes ($\delta^{13}C_{org}$) with contradictory results (Strauss et al., 2015; Weiss et al., 2016). Only few studies have measured $CO_2$ and $CH_4$ production potentials from Siberian Yedoma deposits under laboratory conditions (Dutta et al., 2006; Knoblauch et al., 2013, 2018; Lee et al., 2012; Zimov et al., 2006). In this study, we present incubation data from late Pleistocene Yedoma and Holocene interglacial deposits from three locations in northeast Siberia. We hypothesize that OM deposited during glacial periods experienced little pre-freezing transformation and thus provides a more suitable substrate for future microbial decomposition and greenhouse gas production post-thawing than Holocene deposits.




## 2    Study region and sample material

Three locations in the Laptev Sea region in northeast Siberia were studied (Fig. 1). The whole region is underlain by continuous permafrost reaching depths of 450–700 m onshore and 200–600 m offshore (Romanovskii et al., 2004) with ground temperatures of -11 °C for terrestrial permafrost (Drozdov et al., 2005) and -1 °C for submarine permafrost (Overduin et al., 2015). Long, cold winters and short, cool summers characterize the modern climate. Mean annual (1971–2000) temperatures and precipitation sums are -13.3 °C and 266 mm at the central Laptev Sea coast (Tiksi, WMO station 21824) and -14.9 °C and 145 mm in the eastern Laptev Sea region (Mys Shalaurova, WMO station 21647, Bulygina and Razuvaev, 2012). Modern vegetation cover is dominated by erect dwarf-shrub and in places by sedge, moss, low-shrub wetland vegetation or tussock-sedge, dwarf-shrub, moss tundra vegetation (CAVM Team, 2003). A compilation of the regional stratigraphic scheme used in this work with paleoclimate and vegetation history is summarized in Table 1.

The first study location is on Muostakh Island (71.61° N, 129.96° E), an island in the Buor Khaya Bay 40 km east of Tiksi. Between 1951–2013, the area and volume of Muostakh Island, which is subject to major coastal erosion (up to -17 m a$^{-1}$) and thaw subsidence, decreased by 24% and 40%, respectively (Günther et al., 2015). The entire sedimentary sequence of Muostakh Island (sample code MUO12) was sampled in three vertical sub-profiles on the northeastern shore (Meyer et al., 2015). In the current study, we used 14 sediment samples from the entire MUO12 sequence between 0.5–15.6 meters below surface (mbs), which corresponds to19.5–4.4 meters above sea level (masl).

The second study location is on the Buor Khaya Peninsula (71.42° N, 132.11° E). Thermokarst processes affect 85% of the region, which resulted in >20 m of permafrost subsidence in some areas (Günther et al., 2013). Long-term (1969–2010) coastal erosion rates along the western coast of the Buor Khaya Peninsula are ~-1 m a$^{-1}$ (Günther et al., 2013). On top of the Yedoma hill, approximately 100 m from the cliff edge, a 19.8 m long permafrost core (sample code BK8) was drilled (Grigoriev et al., 2013). Detailed cryolithological, geochemical, and geochronological data (Schirrmeister et al., 2017), palynological analysis (Zimmermann et al., 2017b), and lipid biomarker studies (Stapel et al., 2016) were previously published for the BK8 site. In the current study, 20 sediment samples spread evenly between the surface and 19.8 mbs (or 34 to 14.2 masl) were analyzed, excluding an ice wedge between 3.2–8.5 mbs.

The third sampling location is on Bol'shoy Lyakhovsky Island (73.34° N; 141.33° E), the southernmost island of the New Siberian Archipelago. Four cores (sample code L14) were drilled on the southern coast





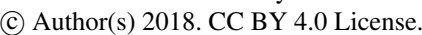

(Schwamborn and Schirrmeister, 2015). Core descriptions as well as pollen and plant DNA analyses can
be found in Zimmermann et al. (2017a), while biomarkers and pore water analysis can be found in Stapel
et al. (2018). Based on previous stratigraphic studies from this location (e.g. Andreev et al., 2009;
Wetterich et al., 2009, 2014) we focused on two cores, which represent the here investigated MIS 1–
MIS 3 period. The first core, L14-05, was recovered from inside a thermokarst basin, 4 km west of the
Zimov'e River mouth, with Holocene thermokarst deposits overlying thawed and refrozen taberal
Yedoma deposits (Wetterich et al., 2009). Five sediment samples between 0–7.9 mbs (11.5–3.6 masl)
were analyzed for the current study. The second core, L14-02, was taken on a Yedoma hill about 1 km
west of the Zimov'e River mouth. The entire core was 20.0 m long, including wedge ice below 10.9 mbs.
Five sediment samples from the top to a depth of 10.9 mbs (32.2–21.3 masl) were incubated for the
current study.

**3    Methods**
**3.1.   Dating**
Radiocarbon dating was performed on plant macro fossils for MUO12 (Meyer et al., unpublished data),
BK8 (Schirrmeister et al., 2017), and L14 samples (Zimmermann et al., 2017a) using the AMS facilities
of University of Poznan and Cologne University. Additionally, feldspars grains from the BK8 core at 12.6–
12.75 mbs, 16.0–16.35 mbs 18.5–18.7 mbs were dated by infrared-stimulated luminescence (IRSL)
(Schirrmeister et al., 2017).

**3.2.   Geochemical characteristics**
Gravimetric water contents were calculated as the weight difference between wet and dried (105 °C)
samples. pH values were measured in a suspension of 5 g thawed sediment in 12.5 ml distilled water
(CG820, Schott AG, Mainz, Germany). For sediment chemical analyses, bulk samples were dried at
70°C and milled. Total carbon (TC) and total nitrogen (TN) contents were measured with an element
analyzer (VarioMAX cube, Elementar Analysensysteme GmbH, Hanau, Germany), while TOC contents
were measured with a liquiTOC II coupled to a solids module (Elementar Analysensysteme GmbH,
Hanau, Germany). The $\delta^{13}C_{org}$-values were measured with an isotope-ratio mass spectrometer (Delta V,
Thermo Scientific, Dreieich, Germany) coupled to an elemental analyzer (Flash 2000, Thermo Scientific,
Dreieich, Germany) after samples were treated with phosphoric acid to release inorganic carbon.





### 3.3. Incubation

Frozen samples were slowly thawed from -18 °C to 4 °C over 48 h in a refrigerator and homogenized. Anaerobic incubations were prepared under a nitrogen atmosphere in a glove box. Approximately 15–30 g thawed sediment was weighed into glass bottles and sealed with rubber stoppers. Anaerobic samples were saturated with 5–20 ml of nitrogen-flushed, $CO_2$-free distilled water and the headspace was exchanged with molecular nitrogen. The headspace of aerobic incubation bottles was exchanged with synthetic air (20% oxygen, 80% nitrogen).

Samples from all three study locations were incubated for 134 incubations days at 4 °C. During this time the headspace $CO_2$ and $CH_4$ concentrations were measured weekly to biweekly. The incubation of samples from the Buor Khaya Peninsula and Bol'shoy Lyakhovsky Island continued until 785 days. The measuring intervals gradually decreased to every 8–12 weeks for the remaining incubation period. Gas concentrations inside each bottle were determined by a gas chromatograph (GC 7890 Agilent Technologies, Santa Clara, USA) equipped with a nickel catalyst to reduce $CO_2$ to $CH_4$ and a flame ionizing detector (FID). Gases were separated on a PorapakQ column with helium as carrier gas. If the headspace concentration of $CO_2$ in aerobic incubation bottles approached 3%, the headspace was again exchanged with synthetic air.

The amount of gas in the headspace was calculated from the concentration in the headspace, headspace volume, incubation temperature, and pressure inside the bottle using the ideal gas law. The amount of gas dissolved in water was calculated from the gas concentration in the headspace, pressure inside the bottle, water content, pH, and gas solubility. Solubility for $CO_2$ and $CH_4$ in water was calculated after Carroll et al. (1991) and Yamamoto et al. (1976), respectively. To account for the dissociation of carbonic acid in water at different pH values, we used dissociation constants from Millero et al. (2007).

### 3.4. Statistics

Differences in mean values were analyzed with the Kruskal-Wallis test followed by multiple post-hoc Mann-Whitney tests with Bonferroni adjustment for multiple group comparisons. We tested for differences between deposits from different periods as well as for differences between deposits from the same period but from different locations. In both cases, the number of post-hoc comparisons was three, giving an adjusted significance level of 0.017. All statistical analyses were performed using MATLAB® (MATLAB and Statistics Toolbox Release 2015b, The MathWorks Inc., Natick, MA, USA).



## 4    Results

### 4.1.  Chronostratigraphy and geochemical characteristics

The sedimentary sequence on Muostakh Island was divided into three sections, which were separated by two erosional contacts and sharply intersecting ice wedges (Meyer et al., 2015). Based on radiocarbon ages (Meyer et al., unpublished data), these sections could be separated into three periods. Deposits from the uppermost section between 0.5–2.4 mbs were classified as Holocene deposits from the MIS 1 and deposits from the late glacial to early Holocene transition, confirmed by radiocarbon ages of 7.5 and 13.2 ka BP for samples at 1.3 and 2.4 mbs, respectively. The middle section between 4–10 mbs yielded radiocarbon ages of 16.1–18.9 ka BP and were therefore classified as Sartan stadial deposits from the MIS 2. The lowermost section between 11.3–15.6 mbs yielded radiocarbon ages of 41.6–45.9 ka BP and represents the MIS 3 Kargin interstadial.

The BK8 core from the Buor Khaya Peninsula was subdivided into four sections. The first section between 0–0.5 mbs represents the seasonally thawed active layer. The subdivision of the permafrost deposits below the active layer was based on previously published radiocarbon and IRSL ages (Schirrmeister et al., 2017). Deposits from the second section between 0.5–3.2 mbs yielded radiocarbon ages between 9.7–11.4 ka BP, which corresponds to the late glacial transition to the early Holocene. The third section between 3.2–8.5 mbs consisted of an ice wedge, which was not sampled for the current study. The fourth section between 8.5–18.9 mbs yielded infinite radiocarbon ages of >50 ka BP. The additional IRSL ages of feldspar grains yielded deposition ages of ~45 ka BP. Thus, sediments from this section were classified as deposits from the Kargin interstadial.

The upper 0.5 m from both cores from Bol'shoy Lyakhovsky Island represent the active layer. Radiocarbon ages of the L14-05 core from the thermokarst basin ranged between 2.2–10.1 ka BP for the upper core section between 0–1.7 mbs and 51.2–54.6 ka BP for deposits below 5.8 mbs (Zimmermann et al., 2017a). Based on these ages, stratigraphic interpretations from a nearby outcrop (Wetterich et al., 2009) and the available palynological data (Zimmermann et al., 2017a), the L14-05 core was divided into two parts. The upper part between 0–5.5 mbs were deposited during the Holocene and late glacial transition, while deposits below 5.5 mbs was deposited during the Kargin interstadial. Deposits from the L14-02 core from the Yedoma hill yielded radiocarbon ages between 33.1–62.8 ka BP, which corresponds to deposition during the MIS3 Kargin interstadial.

Overall, the permafrost deposits showed a wide range in TOC contents (0.8 – 6.3 wt%), C/N (4.6 – 29.4), and $\delta^{13}C_{org}$ (-29.0 – -22.8 ‰VPDB, Fig. 2). Generally higher TOC contents and C/N were found in



deposits from the Holocene and Kargin interstadial than in deposits from the Sartan stadial (Mann-
Whitney test, p < 0.017), while the $\delta^{13}C_{org}$-values were significantly higher in Sartan stadial deposits
(Mann-Whitney test, p < 0.001).

**4.2. Muostakh Island**
Based on the TOC content, $CO_2$ production after 134 incubation days from sediment samples from the
MUO12 sequence ranged between 4.8–60.7 mg $CO_2$-C g$^{-1}$ OC under aerobic conditions and 0.5–20.9
mg $CO_2$-C g$^{-1}$ OC under anaerobic conditions (Fig. 3). Higher aerobic $CO_2$ production was generally
observed in the lowermost Kargin deposits between 11.3–15.6 mbs but elevated $CO_2$ production rates
were also observed at 1.6 mbs, 6 mbs, and 10 mbs. Under anaerobic conditions, the highest production
was observed at 6 mbs (19.3 ± 1.4 mg $CO_2$-C g$^{-1}$ OC), which was nearly twice as high as in most other
samples. No methanogenesis was observed in any Muostakh Island samples over the 134-day
incubation period.

**4.3. Buor Khaya Peninsula**
After 134 incubation days, $CO_2$ production in BK8 core samples ranged between 2.2–64.1 mg $CO_2$-C g$^{-1}$
OC aerobically and 2.2–17.1 mg $CO_2$-C g$^{-1}$ OC anaerobically (Fig. 4), which is within the same range as
production in samples from Muostakh Island over the same incubation period (Fig. 3). The highest
production was observed in the active layer. Production then decreased sharply between 0.5–3.2 mbs
but increased again in Kargin interstadial deposits below the ice-wedge. Methanogenesis was only
observed in the active layer, but in much smaller quantity than anaerobic $CO_2$ (0.37 ± 0.22 mg $CH_4$-C g$^{-1}$
OC compared to 13.3 ± 3.6 mg $CO_2$-C g$^{-1}$ OC).
To assess the long-term decomposability, all BK8 core samples were incubated for a total of 785
days. After 785 incubation days, $CO_2$ production ranged between 4.6–131.1 mg $CO_2$-C g$^{-1}$ OC under
aerobic conditions and 2.2–43.0 mg $CO_2$-C g$^{-1}$ OC under anaerobic conditions. $CO_2$ production rates,
however, decreased sharply within the first weeks of incubation. On average, 58 ± 12% of the aerobically
and 86 ± 24% of the anaerobically produced $CO_2$ after 785 incubation days was already produced within
the first 134 days. In contrast, $CH_4$ production in the active layer increased 30-fold to 11.4 ± 3.0 mg
$CH_4$-C g$^{-1}$ OC. Additionally, two out of three replicates at 10 mbs also showed active methanogenesis
after 785 days (4.0 mg $CH_4$-C g$^{-1}$ OC and 12.7 mg $CH_4$-C g$^{-1}$ OC, respectively).



### 4.4. Bol'shoy Lyakhovsky Island

Aerobic $CO_2$ production after 134 incubation days in samples from the L14 cores ranged between 3.7–18.9 mg $CO_2$-C $g^{-1}$ OC (Fig. Figure **5**). The mean aerobic $CO_2$ production in all Kargin interstadial deposits from Bol'shoy Lyakhovsky Island (9.2 ± 4.7 mg $CO_2$-C $g^{-1}$ OC) was significantly lower (Mann-Whitney test, $p < 0.001$) than $CO_2$ production in MIS 3 deposits from Muostakh Island (32.2 ± 15.6 mg $CO_2$-C $g^{-1}$ OC) and the Buor Khaya Peninsula (26.0 ± 12.6 mg $CO_2$-C $g^{-1}$ OC). Anaerobic $CO_2$ production in Kargin deposits ranged between 3.2–11.6 mg $CO_2$-C $g^{-1}$ OC, which was within the same range as production observed from the other two locations. No $CH_4$ production was observed in any L14 samples after 134 days.

After 785 incubation days, aerobic and anaerobic $CO_2$ production ranged between 11.0–55.2 mg $CO_2$-C $g^{-1}$ OC and 3.0–27.0 mg $CO_2$-C $g^{-1}$ OC, respectively. Active methanogenesis was only observed in 2 out of 3 replicates from the active layer from the L14-05 core (0.41 mg $CH_4$-C $g^{-1}$ OC and 0.63 mg $CH_4$-C $g^{-1}$ OC). $CH_4$ production was therefore an order of magnitude lower than anaerobic $CO_2$ production in the same sample (5.7 mg $CO_2$-C $g^{-1}$ OC and 4.6 mg $CO_2$-C $g^{-1}$ OC) and also an order of magnitude smaller than $CH_4$ production in the active layer from the Buor Khaya Peninsula.

### 4.5. Decomposability of permafrost OM deposited under different climatic regimes

Overall, permafrost OM deposited during the MIS 3 Kargin interstadial supported the highest greenhouse gas production (Fig. Figure **6**). After 134 days of aerobic incubation, 0.2–6.1% of the initially available OC (mean 2.3 ± 1.4%) was decomposed to $CO_2$. This was significantly more (Mann-Whitney test, $p < 0.001$) than in deposits from the Holocene and late glacial transition, where production ranged between 0.4–4.0% (mean 1.2 ± 0.8%). The aerobic $CO_2$ production in MIS 2 Sartan stadial deposits ranged between 0.5–4.2% (mean 1.7 ± 1.2 %). Anaerobically, 3.3 times less $CO_2$ was produced (Pearson correlation coefficient $r = 0.63$, $p < 0.001$). The lowest production was observed in Holocene and late glacial transition deposits, where 0.1–1.1 % of the OC was anaerobically decomposed to $CO_2$ (mean 0.5 ± 0.3). This was significantly less (Mann-Whitney test, $p < 0.01$) than in glacial deposits, where 0.4–2.1% (mean 0.9 ± 0.5%) and 0.2–1.6 % of initial OC (mean 0.7 ± 0.3%) were decomposed in Sartan stadial and Kargin interstadial deposits, respectively.



## 5    Discussion

### 5.1.  Organic matter decomposability

The ice-rich permafrost deposits of Muostakh Island, the Buor Khaya Peninsula, and Bol'shoy Lyakhovsky Island are typical for northeast Siberia and the geochemical characteristics (TOC, C/N, $\delta^{13}C_{org}$) were all within the range of other permafrost deposits in the region (Schirrmeister et al., 2011). We hypothesized, that the climatic conditions during deposition affected the amount and decomposability of preserved OM and thus greenhouse gas production potentials after thaw. OM decomposability in degrading ice-rich permafrost therefore needs to be interpreted against the paleo-environmental background.

One way to analyze the OM source in more detail is sedimentary ancient DNA (sedaDNA), which can be used to reconstruct local plant communities and infer predominant climatic conditions (Willerslev et al., 2004). In the BK8 core, a total of 134 vascular plants and 20 bryophytes were identified (Zimmermann et al., 2017b). *Salix*, Poaceae and Cyperaceae, whose roots are a main OM source in tundra soils (Iversen et al., 2015), are present throughout the core. The taxonomic richness was highest between 8.35–16 mbs, where also high $CO_2$ production was observed. This core section, which belongs to the MIS 3 Kargin interstadial (Schirrmeister et al., 2017), was dominated by swamp and aquatic taxa, pointing towards a water-saturated environment, likely a low-centered ice-wedge polygon (Zimmermann et al., 2017b). Together with higher TOC contents at these depths, this suggests accumulation of relatively undecomposed OM under anaerobic conditions, which can be quickly decomposed after thaw (de Klerk et al., 2011). Furthermore, high concentrations of branched glycerol dialkyl glycerol tetraether (br-GDGT), a microbial membrane compound, are indicative of a soil microbial community, which developed when the climate was relatively warm and wet (Stapel et al., 2016). Overall, br-GDGT concentrations were highest at 10 mbs, 11.2 mbs, and 15 mbs (Stapel et al., 2016), which corresponds to the same levels where the highest $CO_2$ production was observed. In contrast, lower abundance of swamp taxa and higher abundance of terrestrial taxa at 8.8 mbs and >15 mbs (Zimmermann et al., 2017b), suggest that intermittently drier conditions existed. This resulted in accelerated OM decomposition under aerobic conditions prior to OM incorporation into the permafrost, lower TOC contents as well as lower $CO_2$ production potentials at these depths.

Sediments above the ice-wedge in the BK8 core showed similar TOC contents, C/N, and $\delta^{13}C_{org}$-values compared to the rest of the core, but $CO_2$ production was consistently low in this section. This ~3 m long core section yielded radiocarbon ages of 11.4–10.1 ka BP (Schirrmeister et al., 2017), which





corresponds to the late glacial-early Holocene transition. After the Last Glacial Maximum (LGM), temperatures were favorable for increased microbial decomposition of active layer OM, which led to the preservation of comparatively stable OM fractions after the material was incorporated into the permafrost. Similar conclusions can be drawn for Holocene deposits in thermokarst landforms or on top of Yedoma deposits. On the one hand, they received fresh OM inputs, which explains their relatively high TOC contents (Strauss et al., 2015). On the other hand, intensive thermokarst development during the late glacial transition and the early Holocene likely resulted in higher decomposition rates in thawed soils and the loss of labile OM compounds before the sediments were refrozen when climate conditions deteriorated after the Holocene climate optimum. If these sediments were to thaw again in the future, results from the current study suggest that the decomposability of the remaining OM will be comparatively low. Although climatic conditions influence the vegetation composition and OM source on a regional level, the local depositional environment as well as post-depositional processes likely also control the amount and decomposability of the OM that is presently incorporated into permafrost.

First results of *in situ* $CO_2$ fluxes from Muostakh Island were published by Vonk et al. (2012). They estimate that 66% of the thawed out OC from degrading ice-rich permafrost deposits can be decomposed to greenhouse gases and released back to the atmosphere before the material is reburied in the Laptev Sea. However, no further detailed palynological or microbial biomarker studies are yet available for the MUO12 sequence. The closest reference locations is the comprehensive permafrost record at the Mamontovy Khayata section on the Bykovsky Peninsula (Andreev et al., 2002; Sher et al., 2005). Sea level rise after the last glacial period, coastal erosion, and marine ingression of thermokarst basins eventually separated Muostakh Island from the Bykovsky Peninsula (Grosse et al., 2007; Romanovskii et al., 2004). Today, the distance between the northern tip of Muostakh Island and the southern tip of the Bykovsky Peninsula is about 16 km. Between 58–12 ka BP (Schirrmeister et al., 2002), fine-grained material accumulated on the large flat foreland plain of the today Bykovsky Peninsula area that was exposed at a time of lower sea level (Grosse et al., 2007). The distance between Muostakh Island and the Buor Khaya Peninsula is about 80 km. It is therefore likely that the deposition regimes on Muostakh Island and the Buor Khaya Peninsula were similar to the regime at the Bykovsky Peninsula. This conclusion is also supported by similar OM decomposability. After 134 incubation days, the amount of $CO_2$ production did not differ significantly (Mann-Whitney test, $p = 0.339$) between MIS 3 Kargin deposits from Muostakh Island ($3.2 \pm 1.6\%$ of initial OC aerobically and $0.7 \pm 0.6\%$ anaerobically) and the BK8



core (2.7 ± 1.2% aerobically and 0.8 ± 0.3 % anaerobically), which suggests that the deposits formed
under similar conditions.
Under aerobic conditions, $CO_2$ production of MIS 3 deposits from Bol'shoy Lyakhovsky Island in the
eastern Laptev Sea was nearly three times lower (0.9 ± 0.5% of the initial OC after 134 days) than
observed for Muostakh Island and the Buor Khaya Peninsula in the central Laptev Sea. Considerably
lower temperatures and precipitation characterize the modern climate on Bol'shoy Lyakhovsky Island. It
is also likely that regional differences between the eastern and central Laptev Sea region would have
affected the paleo-climate (Anderson and Lozhkin, 2001; Lozhkin and Anderson, 2011; Wetterich et al.,
2011, 2014). Different summer temperatures, precipitation, thaw depth, and vegetation composition
could explain regional differences in OM quantity and decomposability. Interestingly, the differences in
the amount of OC that was aerobically decomposed were mostly due to differences in the initial $CO_2$
production rates. Maximum $CO_2$ production rates during the first weeks of incubation of Muostakh Island
and Buor Khaya deposits were up to four times higher than in deposits from Bol'shoy Lyakhovsky Island.
However, long-term production rates after >130 incubation days did no longer differ considerably
between the different locations (median 23.3 µg $CO_2$-C $g^{-1}$ OC $d^{-1}$). These rates are within the range of
other long-term production rates from Yedoma deposits in northeast Siberia (Dutta et al., 2006;
Knoblauch et al., 2013) and Alaska (Lee et al., 2012). Considering the large slowly decomposing
permafrost OC pool (Schädel et al., 2014), long-term decomposition rates are likely to provide more
reliable projections of future greenhouse gas emissions from degrading permafrost landscapes.
A distinctive feature of the Muostakh Island sequence is the preservation of MIS 2 Sartan deposits,
which are only sparsely preserved in northeast Siberia (Wetterich et al., 2011). Interestingly, mean
aerobic $CO_2$ production in Sartan deposits from Muostakh Island was lower than in Kargin deposits, but
slightly higher under anaerobic conditions, but the difference was not statistically significant (Mann-
Whitney test, $p = 0.205$). The rapid deposition of 8 m thick comparatively coarse-grained material in just
a few thousand years between 20–16 ka BP were unfavorable for the development of a stable land
surface and the establishment of a vegetation cover comparable to the Kargin interstadial or Holocene
periods (Meyer et al., unpublished data). Pollen analysis from the corresponding sections on the
Bykovsky Peninsula (Andreev et al., 2002) and Kurungnakh Island in the Lena River Delta (Schirrmeister
et al., 2008; Wetterich et al., 2008) suggest relatively cold and dry summer conditions during this stadial
with sparse vegetation. Relatively undecomposed OM was quickly buried, before it could be transformed
to greenhouse gases.






## 5.2. Long-term production potentials

Long-term greenhouse gas production measurements after 785 days showed that 51% of the aerobically and 83% of the anaerobically produced $CO_2$ were already produced within the first 134 incubation days, highlighting the non-linearity of OM decomposition dynamics (Knoblauch et al., 2013; Schädel et al., 2014) and the importance of the labile OC pool in short term incubations. Assuming no new input of labile OM (e.g. from modern vegetation), decomposition rates are likely to remain low after the labile pool is depleted. Short-term greenhouse gas production and release from thawing ice-rich permafrost will therefore mainly depend on the size of the labile pool. A synthesis study of several incubations studies from high-latitude soils, including Yedoma deposits, estimated the size of the labile OC pool to be generally <5% of the TOC (Schädel et al., 2014). For Yedoma deposits on nearby Kurungnakh Island in the Lena River delta, Knoblauch et al. (2013) estimated the size of the labile pool to be even smaller (<2%).

## 5.3. Methanogenesis

$CH_4$ production from Yedoma deposits, or the lack thereof, is a highly controversial topic in permafrost research (Knoblauch et al., 2018; Rivkina et al., 1998; Treat et al., 2015). In the current work, active methanogenesis was only observed in the active layer and 2 out of 38 Yedoma samples from the BK8 core, but only after a long lag-phase. Within 134 incubation days, no samples from Muostakh Island produced any $CH_4$. . In those samples showing active methanogenesis, $CH_4$ production continued to rise over the 785 incubation days, which is in contrast to anaerobic $CO_2$ production, which decreased with increasing incubation time. Rising $CH_4$ production rates indicate that methanogenic communities still grow in these samples and were not limited by substrate supply. Chemical pore water and bulk sediment analyses from the BK8 core showed that there are high concentrations of both free and OM-bound acetate present in Yedoma deposits, indicating a high substrate potential for methanogenesis (Stapel et al., 2016). Knoblauch et al. (2018) showed that the small contribution of methanogenesis to overall anaerobic permafrost OM decomposition found in short-term incubation studies (Treat et al., 2015) is due to the absence of an active methanogenic community. On a multi-annual timescale, methanogenic communities become active and equal amounts of $CO_2$ and $CH_4$ are produced from permafrost OM under anaerobic conditions. Under future climate warming and renewed thermokarst activity, high levels of $CH_4$ production can locally be expected but depend on favorable conditions such as above-zero




temperatures and anaerobic conditions. It can be expected that the development of an active methanogenic community e.g. by growth or downward migration of modern methanogenic organisms will lead to elevated long-term $CH_4$ production in these deposits (Knoblauch et al., 2018).

## 6    Conclusion

In this study, we investigated greenhouse gas production potentials in degrading ice-rich permafrost deposits from three locations in northeast Siberia. It could be shown that Yedoma deposits generally contained more labile OM than Holocene deposits. However, in addition to the regional climate conditions at the time of OM deposition, local depositional environments also influenced the amount and decomposability of the preserved fossil OM. Within the deposits of the MIS 3 Kargin interstadial, sediments deposited under wet and possibly anaerobic conditions produced more $CO_2$ than sediments deposited under drier aerobic conditions. It is therefore likely, that OM decomposability of the vast Yedoma landscape cannot be generalized solely based on the stratigraphic position. Furthermore, it is expected that $CH_4$ production will play a more prominent role after active methanogenic communities have established since abundant substrates for methanogenesis were present.

**Data availability**

https://www.pangaea.de/ (follows after acceptance and includes all shown datasets)

**Author contributions**

JW and CK designed the study. TO collected sediment samples on Muostakh Island and LS collected cores from the Buor Khaya Peninsula and Bol'shoy Lyakhovsky Island. JW and RT performed the laboratory analyses, with guidance from CK and EMP. JW performed data analyses and wrote the manuscript with contributions from all authors.

**Acknowledgements**

This research was supported by the German Ministry of Education and Research as part of projects CarboPerm (grant no. 03G0836A, 03G0836B) and KoPf (grant no. 03F0764A). We acknowledge the financial support through the German Research Foundation (DFG) to EMP and CK through the Cluster of Excellence "CliSAP" (EXC177), University Hamburg and to TO (grant OP 217/3-1). We also thank the Russian and German participants of the drilling and sampling expeditions, especially Mikhail N. Grigoriev



(Melnikov Permafrost Institute, Yakutsk), Hanno Meyer and Pier Paul Overduin (both Alfred-Wegener-Institute, Potsdam). Additional thanks go to Georg Schwamborn (Alfred-Wegener-Institute, Potsdam) for his assistance with core subsampling and Birgit Schwinge (Institute of Soil Science, Hamburg) for her help in the laboratory.

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





## Figures

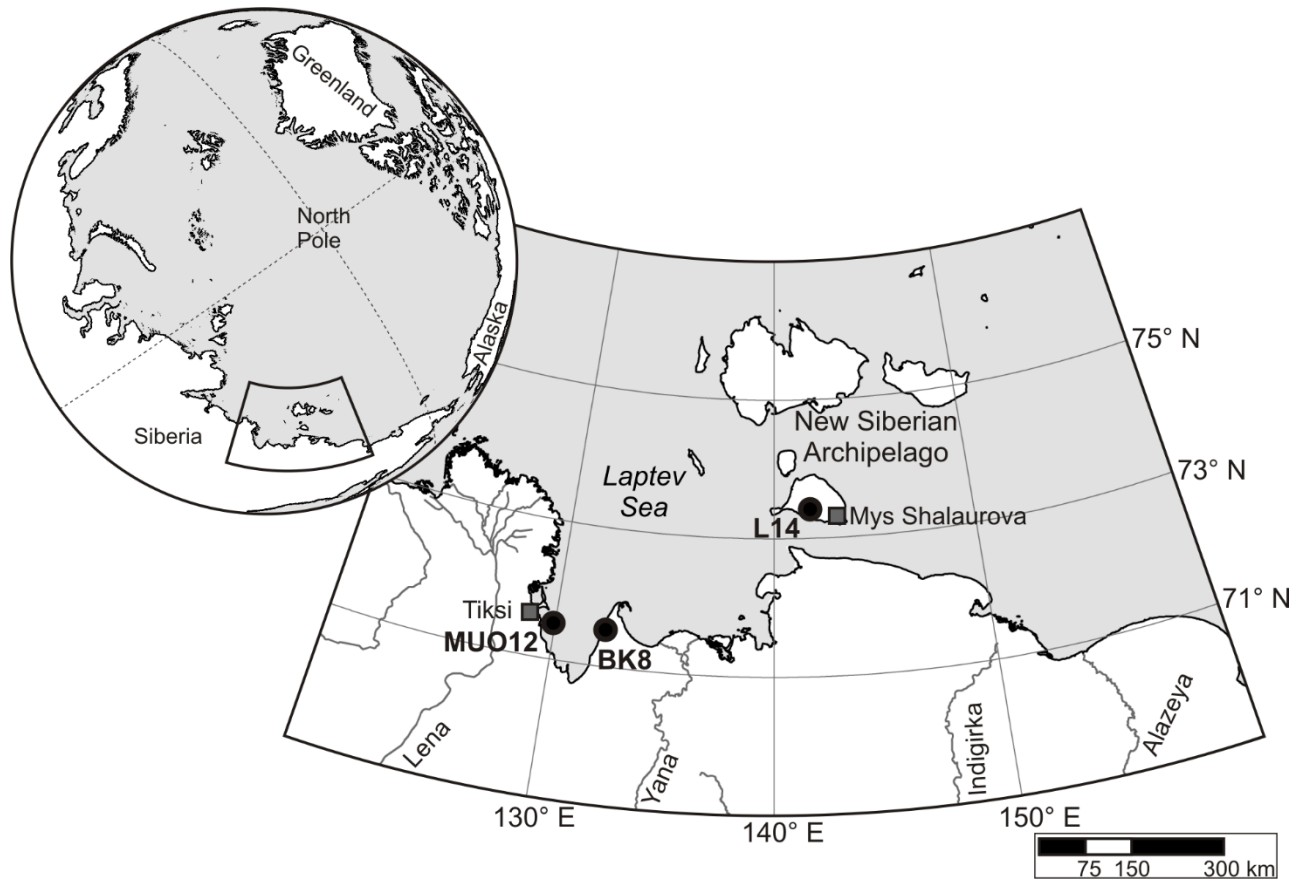

Figure 1. Overview map of the Laptev Sea region with the study locations at Muostakh Island (sample code MUO12), the Buor Khaya Peninsula (BK8) and Bol'shoy Lyakhovsky Island (L14).

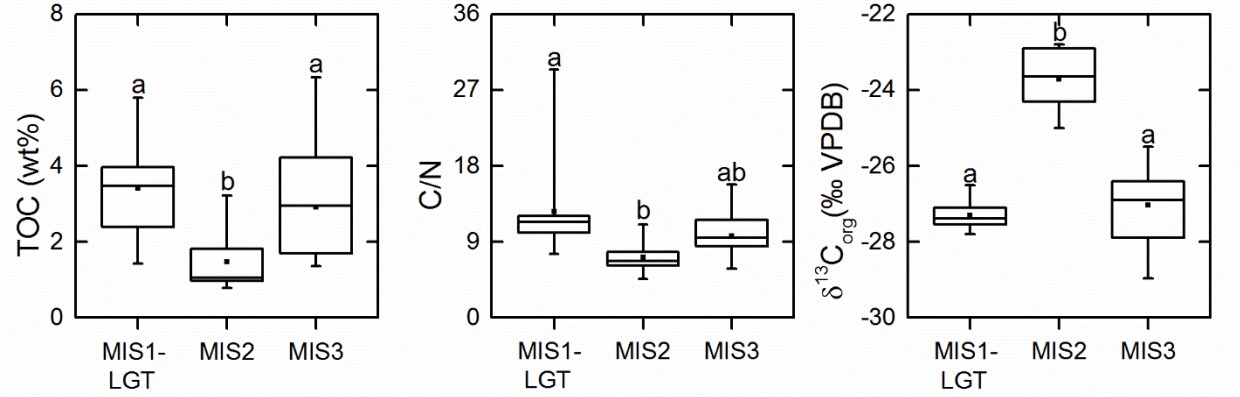

Figure 2. Boxplot of total organic carbon (TOC), total organic carbon to total nitrogen ratio (C/N) and $\delta^{13}C_{org}$-values of permafrost deposits from the MUO12 sequence, the BK8 core, and the two L14 cores



from the Holocene interglacial (MIS 1), including the late glacial transition (LGT) (n = 12), the Sartan
stadial (MIS 2) (n = 6), and the Kargin interstadial (MIS 3) (n = 27). The whiskers show the data range
and the box indicates the interquartile range. The vertical line and square inside the boxes show the
median and mean, respectively. The letters above the whiskers indicate statistically significant
differences in geochemical characteristics between the deposits of different ages (Mann-Whitney test, p
< 0.016 for TOC and C/N, p < 0.001 for $\delta^{13}C_{org}$)

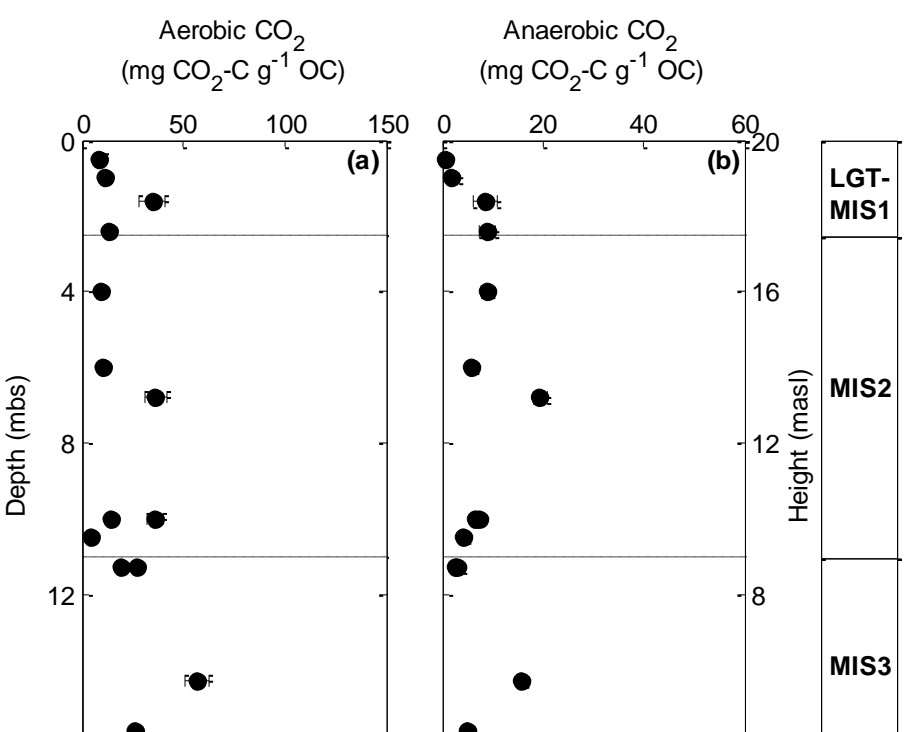

Figure 3. Depth profiles of total aerobic (a) and anaerobic (b) $CO_2$ production per gram organic carbon
($g^{-1}$ OC) in sediment samples from the MUO12 sequence after 134 incubation days at 4 °C for deposits
from the Holocene interglacial (MIS 1), including the late glacial transition (LGT), the Sartan stadial (MIS
2), and the Kargin interstadial (MIS 3). Data are mean values (n = 3) and error bars represent one
standard deviation. Note the different scales. No $CH_4$ production was observed during the 134-days
incubation period.



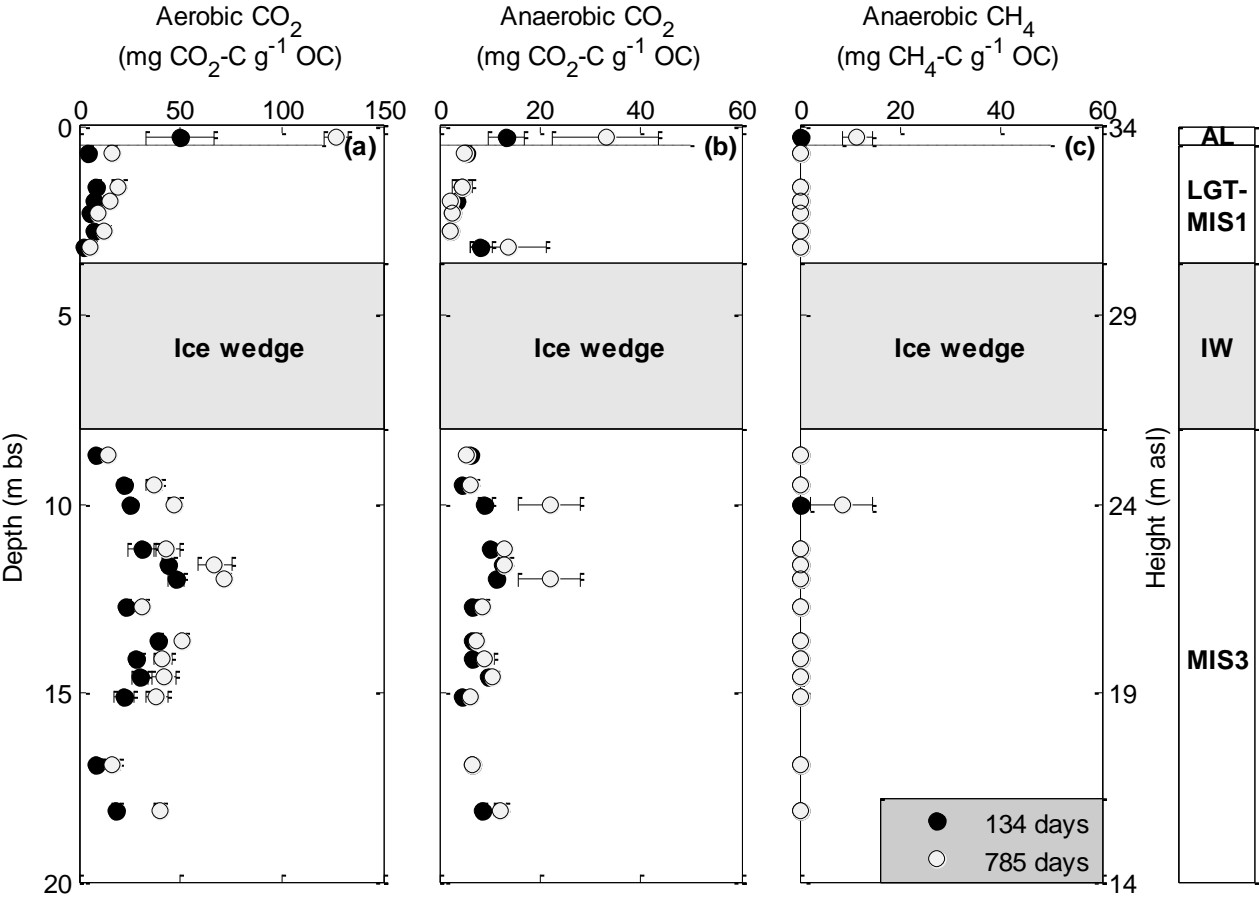

Figure 4. Depth profiles of total aerobic $CO_2$ (a), anaerobic $CO_2$ (b) and anaerobic $CH_4$ (c) production per gram organic carbon (g$^{-1}$ OC) in sediment samples from the BK8 core after 134 (closed symbols) and 785 incubation days (open symbols) at 4 °C for the active layer (AL) and permafrost deposits from the Holocene interglacial (MIS 1), including the late glacial transition (LGT) and the Kargin interstadial (MIS 3). Data are mean values (n = 3) and error bars represent one standard deviation. Note the different scales.





Figure 5. Depth profiles of total aerobic $CO_2$ (a), anaerobic $CO_2$ (b) and anaerobic $CH_4$ (c) production per gram organic carbon (g$^{-1}$ OC) in sediment samples from the L14-05 (a,b,c), and L14-02 cores (d,e,f) after 134 (closed symbols) and 785 incubation days (open symbols) at 4 °C for the active layer (AL) and permafrost deposits from the Holocene interglacial (MIS 1) and the Kargin interstadial (MIS 3). Data are mean values (n = 3) and error bars represent one standard deviation. Note the different scales.





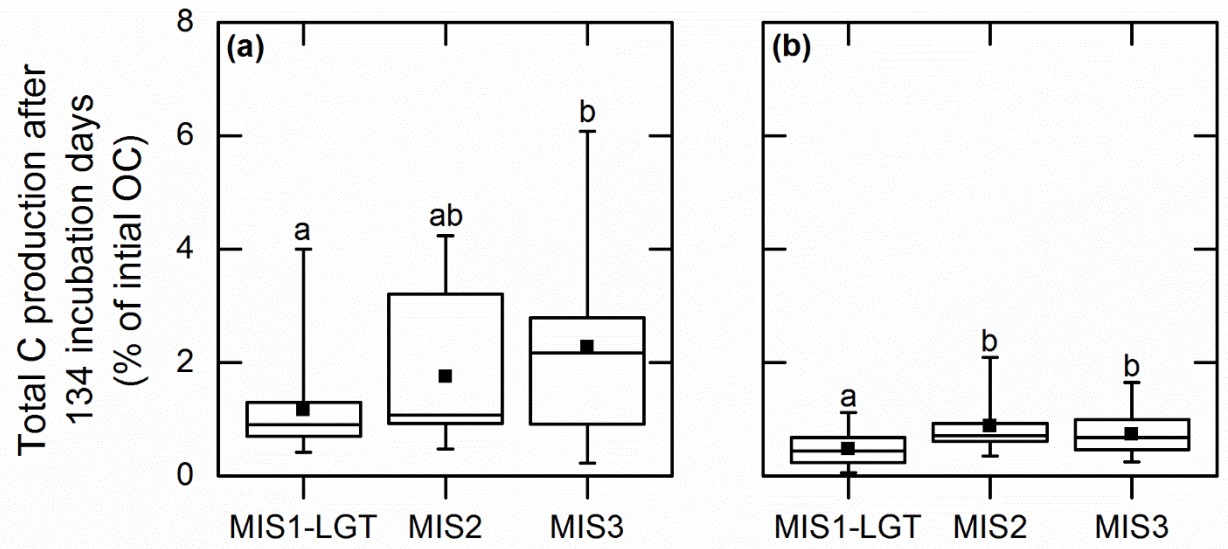

699

Figure 6. Total aerobic (a) and anaerobic (b) $CO_2$-C production after 134 incubation days from permafrost deposits from the MUO12 sequence, the BK8 core, and the two L14 cores from the Holocene interglacial (MIS 1), including the late glacial transition (LGT) (n = 22), the Sartan stadial (MIS 2) (n = 15), and the Kargin interstadial (MIS 3) (n = 50). The whiskers show the data range and the box indicates the interquartile range. The vertical line and square inside the boxes show the median and mean, respectively. The different letters indicate significant differences (Mann-Whitney test, p < 0.016) between deposits from different periods.

707





## Tables

Table 1. Compilation of the regional chronostratigraphy of the Laptev Sea region used in this work with paleoclimate (summer) and vegetation history based on an overview by Andreev et al. (2011) and references therein.

| Age ka BP | Period | Regional chrono-stratigraphy | Marine isotope stage (MIS) | Regional climate and vegetation |
|---|---|---|---|---|
| <10.3 | Holocene | Holocene | MIS 1 | Climate amelioration during the early Holocene; shrub-tundra vegetation gradually disappeared ca 7.6 ka BP |
| ca 10.3–13 | Late glacial-early Holocene transition | | | Climate amelioration after the Last Glacial Maximum; transition to shrubby tundra vegetation |
| ca 13–30 | Late Weichselian glacial (stadial) | Sartan | MIS 2 | Cold and dry summer conditions, winter colder than today; open tundra steppe |
| ca 30–55 | Middle Weichselian glacial (interstadial) | Kargin | MIS 3 | Relatively warm and wet summers; open herb and shrub dominated vegetation |