# Peer review of "Greenhouse gas production in degrading ice-rich permafrost deposits in northeast Siberia"

_Biogeosciences, 2018_

## Referee Comment (RC1) · Anonymous Referee #1 · 20 Jun 2018

General comments

The article "Greenhouse gas production in degrading ice-rich permafrost deposits in northeast Siberia" by Josefine Walz et al. discusses the important issue of permafrost aggradation history and organic matter quality on greenhouse gas ($CO_2$ and $CH_4$) production from degrading yedoma deposits. The findings are based on short-term (134 days) and longer-term (785 days) incubation of samples collected at three locations, and the measured $CO_2$ and $CH_4$ production is linked to a wide array of measurements on geochemical characteristics and the stratigraphy of soil/sediment cores.

The potential future C release from thawing permafrost soils, especially yedoma, is

connected to large uncertainties. Only a limited number of studies address this topic and I particularly value the authors' efforts to asses the longer-term production potential. This topic clearly is of interest to the broader scientific community and I thus consider this manuscript highly relevant for the journal. The manuscript is carefully written, however, it would benefit from some streamlining, especially of the results and discussion section as outlined in my comments below, in order to further improve readability and scientific value of the manuscript.

Specific comments

1) First of all, the results section is rather detailed, partly repeating values presented as figures and reporting many numbers, making it difficult to follow. I would recommend providing part of the information as tables, e.g. an overview table with site names, site codes, ages, mean $CO_2$ and $CH_4$ production rates etc., helping the reader to get an overview of the differences between the three locations. Chapters 4.2, 4.3, 4.4 should be presented under a sub-heading, e.g. "greenhouse gas production". To improve readability, I would further recommend dropping the numbers in the site codes, e.g. just MUO, BK, and L instead of MUO12, BK8, L14.

2) the conclusions drawn in the discussions are partly based on results obtained in this study, but also quite heavily rely on detailed analyses reported in previously published literature (radiocarbon age, plant macrofossils and soil microbial analyses), e.g. L285-303. I suggest to emphasize results measured within this study throughout the discussion. Additionally, section 5.1 of the discussion is rather lengthy and would benefit from some streamlining to more clearly emphasize the main results from this study.

3) The $CO_2$ and $CH_4$ production potential was assessed using separately incubated soil samples, excluding effects of vegetation (e.g. input of fresh OM to the soil system, atmospheric $CO_2$ uptake), and processes among different layers in the soil profile (e.g. diffusion and leaching, as well as priming of old OM). How would the authors relate the gas production measured in these soil incubations to gas release to the atmosphere

under in situ conditions? I would appreciate some more discussion on this part.

Technical corrections / line edits

Abstract L31: if more than 80% were produced during the first 134 of the long-term incubation, shouldn't that rather highlight the importance of the labile C pool, rather than the slowly decomposing C pool?

Introduction L40: give depth range for C stocks, is it 0-3m? L44: "The changes" is slightly vague, please specify. L54: what about MIS 4 and 6, are they not preserved in this region? L60: Either separate sentence by colon ":" or add reference. L62-64: Are those C stocks representative of the whole yedoma deposits, or a certain depth range? L66: Consider replacing "thawed out", e.g. "exposed by degradation of ice-rich permafrost" L67: decomposed to the greenhouse gases carbon dioxide ($CO_2$) and methane ($CH_4$) L77: Do they authors mean "permafrost aggradation"?

Methods L97: Consider replacing "modern" with "current" L155: Might the low temperatures during storage (-18oC) have had an effect on soil microbial community functioning during the incubations? -11oC seems to be the minimum naturally occurring permafrost temperature in this region. L164/165: Some specifications about the gas sampling would be useful, e.g. how many mL of gas were sampled from the headspace for GC analysis? Did gas sampling cause underpressure in the headspace? L171-174: Was the temperature dependency of gas solubility taken into account? I would suggest to provide some more details on the solubility/temperature coefficients used for $CO_2$ and $CH_4$.

Results L236: "anaerobic $CO_2$ production"? L243: increased 30-fold over what time frame?

Discussion L282-284: This seems like an overall conclusion of the study and does not belong in the opening paragraph of the discussion L369-372: Using the term "long-term" for a period of ca. 2 years is slightly questionable, I advise some caution with the

use of this term throughout the manuscript.

Figures L675 (Fig. 3): adding both y-axes (height and depth) to each figure panel, as well as using the same x-axis scaling (e.g. 0-80?) would improve readability of the figure. Since CH4 production is included as a third panel for the other cores, please mention in figure caption why it is not included here. L684 (Fig. 4 and Fig. 5): Please mention AL thickness in figure caption.

---

## Referee Comment (RC2) · Anonymous Referee #2 · 25 Jul 2018

Authors using the word "glacial" very often do not specify that they mean age but not the origin of the deposits they studied. It might confuse the readers who are not familiar with the paleoenvironmental conditions of the area of investigation. I suggest to use marine isotopic stages or regional stratigra[hic units. Lines 64 and 65. I would recommend to authors include in the review of the assessments of the carbon pools in different stratigraphic horizons research published by Shmelev et al (Shmelev, D., Veremeeva, A., Kraev, G., Kholodov, A., Spencer, R. G., Walker, W. S., & Rivkina, E. (2017). Estimation and Sensitivity of Carbon Storage in Permafrost of North‐Eastern Yakutia. Permafrost and Periglacial Processes, 28(2), 379-390.). Line 95. It is not so important for this study, but permafrost temperature in this region varies with the to-

pographic forms and consist of -9 within the thermokarst depressions and -10.5 at the yedoma hills (Kholodov, A., Gilichinsky, D., Ostroumov, V., Sorokovikov, V., Abramov, A., Davydov, S., & Romanovsky, V. (2012, June). Regional and local variability of modern natural changes in permafrost temperature in the Yakutian coastal lowlands, Northeastern Siberia. In Proceedings of the Tenth International Conference on Permafrost, Salekhard, Yamal-Nenets Autonomous District, Russia (pp. 25-29).) For the Results section, I also recommend authors to insert the graphs of dynamics of the greenhouse gases production during the experiment to give readers a better idea about dynamics of the process of organic matter decay.

—————————————————————

---

## Author Comment (AC1) · 14 Aug 2018

We sincerely thank the reviewer for the comments, which helped us to improve our manuscript. Please find the comments (blue) and our reply (black) below.

General comments
The article "Greenhouse gas production in degrading ice-rich permafrost deposits in northeast Siberia" by Josefine Walz et al. discusses the important issue of permafrost aggradation history and organic matter quality on greenhouse gas ($CO_2$ and $CH_4$) production from degrading yedoma deposits. The findings are based on short-term (134 days) and longer-term (785 days) incubation of samples collected at three locations, and the measured $CO_2$ and $CH_4$ production is linked to a wide array of measurements on geochemical characteristics and the stratigraphy of soil/sediment cores.

The potential future C release from thawing permafrost soils, especially yedoma, is connected to large uncertainties. Only a limited number of studies address this topic and I particularly value the authors' efforts to asses the longer-term production potential. This topic clearly is of interest to the broader scientific community and I thus consider this manuscript highly relevant for the journal. The manuscript is carefully written, however, it would benefit from some streamlining, especially of the results and discussion section as outlined in my comments below, in order to further improve readability and scientific value of the manuscript.

Specific comments
1) First of all, the results section is rather detailed, partly repeating values presented as figures and reporting many numbers, making it difficult to follow. I would recommend providing part of the information as tables, e.g. an overview table with site names, site codes, ages, mean $CO_2$ and $CH_4$ production rates etc., helping the reader to get an overview of the differences between the three locations.
We added Tabel 2 as a summary table as suggested and removed some of the detailed numbers from the results section to improve readability.
Chapters 4.2, 4.3, 4.4 should be presented under a sub-heading, e.g. "greenhouse gas production".
Chapters 4.2, 4.3, 4.4 are now presented under the sub-heading, "Greenhouse gas production potentials" as suggested.
To improve readability, I would further recommend dropping the numbers in the site codes, e.g. just MUO, BK, and L instead of MUO12, BK8, L14.
We decided to keep the numbers in the site codes, because the same numbers and codes were used for the same sample material in other studies that are referenced in this study. Hence, keeping codes and numbers facilitates the comparison of data from different studies.

2) the conclusions drawn in the discussions are partly based on results obtained in this study, but also quite heavily rely on detailed analyses reported in previously published literature (radiocarbon age, plant macrofossils and soil microbial analyses), e.g. L285-303. I suggest to emphasize results measured within this study throughout the discussion. Additionally, section 5.1 of the discussion is rather lengthy and would benefit from some streamlining to more clearly emphasize the main results from this study.
We rewrote parts of Section 5.1 and 5.2 to shorten and streamline the discussion and to put more emphasis on the incubation results of this study, see. lines 285–299, lines 319–320, or 359–363.

3) The $CO_2$ and $CH_4$ production potential was assessed using separately incubated soil samples, excluding effects of vegetation (e.g. input of fresh OM to the soil system, atmospheric $CO_2$ uptake), and processes among different layers in the soil profile (e.g. diffusion and leaching, as well as priming of old OM). How would the authors relate the gas production measured in these soil incubations to gas release to the atmosphere under in situ conditions? I would appreciate some more discussion on this part.

Post-thaw processes and the introduction of fresh organic are important points. We included some discussion of the priming effects on organic matter decomposability in section 5.1, L307-312.

Technical corrections / line edits
Abstract L31: if more than 80% were produced during the first 134 of the long-term incubation, shouldn't that rather highlight the importance of the labile C pool, rather than the slowly decomposing C pool?
Yes, the labile pool is important for the production in the initial incubation. But this pool is very small. So over longer time scales, the slowly decomposing C pool will become relevant. In the abstract we now refer to the non-linearity of the decomposition processes and discuss the importance of fast vs slowly decomposing C in the discussion.

Introduction L40: give depth range for C stocks, is it 0-3m?
This is the combined C stock of soils, refrozen thermokarst and Holocene cover deposits in the top 3 m as well as sediments and deltaic deposits below 3 m. We added this information in the text for clarity.
L44: "The changes" is slightly vague, please specify.
We replaced "The changes" with "The effects of elevated atmospheric greenhouse gas concentrations and temperatures on processes in soils and sediments".
L54: what about MIS 4 and 6, are they not preserved in this region?
MIS4 deposits are preserved in the region, but not all MIS 4 deposits are ice complex deposits, e.g. in the Kuchchugui Suite on Bol'shoy Lyakhovsky. In the text, we added the information, that "at some locations the accumulation of Yedoma material may have already started between 80 and 60 ka BP, i.e. during MIS 4" (L 55–56). Dating constraints of older, non-yedoma ice complexes, make it difficult to differentiate if the deposition occurred during the MIS7 or early MIS 6. In the text, we added the possibility of ice complex formation during both late MIS 7/early MIS 6 and MIS 5 (L57-59).

L60: Either separate sentence by colon ":" or add reference.
We added a reference.
L62-64: Are those C stocks representative of the whole yedoma deposits, or a certain depth range?
Those C stocks are for the whole Yedoma domain. We clarified this in the text.
L66: Consider replacing "thawed out", e.g. "exposed by degradation of ice-rich permafrost"
We changed "is thawed out from degrading ice-rich permafrost deposits" to "will be exposed by degradation of ice-rich permafrost" as suggested.
L67: decomposed to the greenhouse gases carbon dioxide (CO2) and methane (CH4)
We changed the sentence as suggested.
L77: Do they authors mean "permafrost aggradation"?
Yes. We corrected this typo.

Methods L97: Consider replacing "modern" with "current"
We replaced "modern" with "current" as suggested.
L155: Might the low temperatures during storage (-18oC) have had an effect on soil microbial community functioning during the incubations? -11oC seems to be the minimum naturally occurring permafrost temperature in this region.
We do not expect that the storage temperature will considerably affect the soil microbial community functioning because -11 °C is the ground temperature at the level of zero amplitude, which is at about 20 meters depth. Above that point, temperatures will be lower in winter, with colder temperatures in the upper permafrost and the coldest temperatures in the active layer reaching values of below -30°C.
L164/165:
Some specifications about the gas sampling would be useful, e.g. how many mL of gas were sampled from the headspace for GC analysis? Did gas sampling cause underpressure in the headspace?

We always worked with slight overpressure. In individual cases of underpressure, which occasionally occurred in the longer incubations, we added 5-10 mL of N2 gas to reestablish overpressure in the bottle. We removed 1 mL of headspace gas for each individual GC measurement and corrected for the cumulative removed gas during sampling. We added this information in the Section 3.3.

L171-174: Was the temperature dependency of gas solubility taken into account? I would suggest to provide some more details on the solubility/temperature coefficients used for CO2 and CH4.

Carroll et al. (1991) and Yamamoto et al. (1976) provide temperature-dependent solubility for CO2 and CH4, respectively. In the text, we added the information "Solubility for CO2 and CH4 in water at 4 °C".

Results L236: "anaerobic CO2 production"?
Yes. We added "production"
L243: increased 30-fold over what time frame?
We added the "between 134 and 785 incubation days".

Discussion L282-284: This seems like an overall conclusion of the study and does not belong in the opening paragraph of the discussion
We moved this sentence to the conclusion section and replaced it here with a more appropriate introductory sentence.
L369-372: Using the term "longterm" for a period of ca. 2 years is slightly questionable, I advise some caution with the use of this term throughout the manuscript.
Were appropriate, we replaced "long-term" throughout the text.

Figures L675 (Fig. 3): adding both y-axes (height and depth) to each figure panel, as well as using the same x-axis scaling (e.g. 0-80?) would improve readability of the figure. Since CH4 production is included as a third panel for the other cores, please mention in figure caption why it is not included here. L684 (Fig. 4 and Fig. 5): Please mention AL thickness in figure caption.
We tried adding both y-axes to each figure panel in Fig. 3, 4, and 5, but this made the figures overly busy, so we decided to keep the y-axes as is. We added the AL thickness in figure captions 4 and 5.

Cited references:
Carroll, J. J., Slupsky, J. D. and Mather, A. E.: The solubility of carbon dioxide in water at low pressure, J. Phys. Chem. Ref. Data, 20(6), 1201, doi:10.1063/1.555900, 1991.
Yamamoto, S., Alcauskas, J. B. and Crozier, T. E.: Solubility of methane in distilled water and seawater, J. Chem. Eng. Data, 21(1), 78–80, doi:10.1021/je60068a029, 1976.

---

## Author Comment (AC2) · 14 Aug 2018

Thank you for the helpful suggestions on this manuscript. Please find the comments (blue) and our reply (black) below.

Authors using the word "glacial" very often do not specify that they mean age but not the origin of the deposits they studied. It might confuse the readers who are not familiar with the paleoenvironmental conditions of the area of investigation. I suggest to use marine isotopic stages or regional stratigraphic units.
Where necessary, we changed "glacial" to refer only to age and not origin to avoid ambiguity, e.g. line 23 and line 274.

Lines 64 and 65. I would recommend to authors include in the review of the assessments of the carbon pools in different stratigraphic horizons research published by Shmelev et al (Shmelev, D.,Veremeeva, A., Kraev, G., Kholodov, A., Spencer, R. G., Walker, W. S., & Rivkina, E. (2017). Estimation and Sensitivity of Carbon Storage in Permafrost of North-Eastern Yakutia. Permafrost and Periglacial Processes, 28(2), 379-390.).
We included this information as suggested in lines 69–72.

Line 95. It is not so important for this study, but permafrost temperature in this region varies with the topographic forms and consist of -9 within the thermokarst depressions and -10.5 at the yedoma hills (Kholodov, A., Gilichinsky, D., Ostroumov, V., Sorokovikov, V., Abramov, A., Davydov, S., & Romanovsky, V. (2012, June). Regional and local variability of modern natural changes in permafrost temperature in the Yakutian coastal lowlands, Northeastern Siberia. In Proceedings of the Tenth International Conference on Permafrost, Salekhard, Yamal-Nenets Autonomous District, Russia (pp. 25-29).)
Thank you for this reference. However, we agree that the temperature differences between topographic forms are not of utmost importance for the current study.

For the Results section, I also recommend authors to insert the graphs of dynamics of the greenhouse gases production during the experiment to give readers a better idea about dynamics of the process of organic matter decay.
Thank you for this important comment. We will deposit all the data of the current manuscript on PANGAEA (https://doi.pangaea.de/10.1594/PANGAEA.892950) so we decided not to insert the graphs of dynamics of the greenhouse gas production during the incubation of the samples in the manuscript. Since we incubated 117 individual samples, this cannot be done in a reasonable way.